# Rescue Potential of Supportive Embryo Culture Conditions on Bovine Embryos Derived from Metabolically Compromised Oocytes

**DOI:** 10.3390/ijms21218206

**Published:** 2020-11-02

**Authors:** Anouk Smits, Jo L. M. R. Leroy, Peter E. J. Bols, Jessie De Bie, Waleed F. A. Marei

**Affiliations:** 1Gamete Research Centre, Laboratory for Veterinary Physiology and Biochemistry, Department of Veterinary Sciences, University of Antwerp, 2610 Wilrijk, Belgium; jo.leroy@uantwerpen.be (J.L.M.R.L.); peter.bols@uantwerpen.be (P.E.J.B.); jessie.debie@uantwerpen.be (J.D.B.); waleed.marei@uantwerpen.be (W.F.A.M.); 2Centre for the Evaluation of Vaccination, Vaccine and Infectious Disease Institute, University of Antwerp, 2610 Wilrijk, Belgium; 3Department of Theriogenology, Faculty of Veterinary Medicine, Cairo University, Giza 12211, Egypt

**Keywords:** free fatty acids, fertility, in vitro culture supplementation

## Abstract

Elevated non-esterified fatty acid (NEFA), predominantly palmitic acid (PA), concentrations in blood and follicular fluid are a common feature in maternal metabolic disorders such as obesity. This has a direct negative impact on oocyte developmental competence and the resulting blastocyst quality. We use NEFA-exposure during bovine oocyte in vitro maturation (IVM) as a model to mimic oocyte maturation under maternal metabolic stress conditions. However, the impact of supportive embryo culture conditions on these metabolically compromised zygotes are not known yet. We investigated if the addition of anti-apoptotic, antioxidative and mitogenic factors (namely, Insulin-Transferrin-Selenium (ITS) or serum) to embryo culture media would rescue development and important embryo quality parameters (cell proliferation, apoptosis, cellular metabolism and gene expression patterns) of bovine embryos derived from high PA- or high NEFA-exposed oocytes when compared to controls (exposed to basal NEFA concentrations). ITS supplementation during in vitro culture of PA-exposed oocytes supported the development of lower quality embryos during earlier development. However, surviving blastocysts were of inferior quality. In contrast, addition of serum to the culture medium did not improve developmental competence of PA-exposed oocytes. Furthermore, surviving embryos displayed higher apoptotic cell indices and an aberrant cellular metabolism. We conclude that some supportive embryo culture supplements like ITS and serum may increase IVF success rates of metabolically compromised oocytes but this may increase the risk of reduced embryo quality and may thus have other long-term consequences.

## 1. Introduction

Maternal metabolic disorders like obesity are known to affect reproductive physiology, ultimately leading to a disappointing fertility. Incidences of these metabolic health disorders are dramatically increasing worldwide and have been strongly linked to a significant loss of reproductive capacity [1,2].

Maternal metabolic disorders are associated with upregulated lipolysis and elevated non-esterified fatty acid (NEFA) concentrations in the blood and in the ovarian follicular fluid (FF) [3,4,5,6]. It has been shown that elevated NEFA concentrations in the FF directly affects oocyte developmental capacity [4], and has been associated with poor cumulus-oocyte-complex morphology in humans [7]. Pasquali et al. [8] reported a significant reduction in oocyte quality of obese patients. When FF of obese patients was added during bovine in vitro oocyte maturation (IVM), oocyte developmental competence and embryo quality were negatively affected [9]. The predominant NEFAs that increase during metabolic disorders are the lipotoxic saturated palmitic (PA; 16:0) and stearic (SA; 18:0) acids and the mono-unsaturated oleic (OA; 18:1) acid [10]. A retrospective study demonstrated that pregnancy results after intracytoplasmic sperm injection (ICSI) were negatively associated with the PA concentration in the FF upon oocyte pick-up [11]. Furthermore, the increased concentration of PA in human FF was linked to lower oocyte nuclear maturation and embryo cleavage rates [12]. In animal models, exposure to a mixture of predominant NEFAs (PA, SA and OA) or to PA only at pathophysiological concentrations during IVM increased apoptosis in cumulus cells [4], altered expression of genes related to oxidative stress in oocytes, reduced embryo development to the blastocyst stage, reduced blastocyst cell numbers and increased blastocyst apoptotic cell indices compared to NEFA-free controls [13,14,15,16].

There is a growing evidence that oxidative stress plays a key role in the pathogenesis of reduced oocyte developmental competence under lipotoxic conditions. The affected oocytes have higher reactive oxygen species (ROS) content and exhibit upregulation of reduction-oxidation (REDOX)-related genes [16,17,18]. Importantly, it has been demonstrated that embryos derived from PA-exposed oocytes exhibited persistent oxidative stress and loss of mitochondrial activity despite in vitro culture in standard, fatty acid-free conditions [19]. In addition, the produced blastocysts, resulting from high NEFA-exposed oocytes, displayed a less active oxidative metabolism, altered DNA methylation and transcriptomic fingerprints compared to blastocysts exposed to basal NEFA concentrations during IVM [15,20]. Together, these data illustrate the negative impact of exposure to lipotoxic conditions during maturation on early embryo development and on embryo quality [21,22]. 

On the other hand, the microenvironment in which early embryos grow also has a significant impact on developmental rate and quality. Rizos et al. [23] stated that the in vitro culture (IVC) system is a major determinant of blastocyst quality, whereas the developmental capacity itself is mainly determined during maturation. Nevertheless, it was illustrated that transferring in vitro produced embryos to sheep oviducts yielded significantly higher rates of embryonic development compared to those cultured in simple in vitro conditions [24]. In humans, early transfer of zygotes to the uterus could rescue embryo development and improve pregnancy and birth rates in patients with recurrent production of highly fragmented in vitro fertilization (IVF) embryos [25]. 

Routine in vitro embryo production (IVP) protocols use culture media (such as synthetic oviductal fluid (SOF)) that provide a supportive microenvironment during early embryo development. This can be beneficial as it may provide a recovery potential to the growing embryo originating from a low-quality oocyte. This suggests that supportive embryo culture conditions, for example, enriched with antioxidants, may be used to reduce oxidative stress and prevent aggravation of cell damage during in vitro embryo development when oocytes are collected from metabolically compromised obese patients. 

Culture media are often supplemented with serum which is known to improve oocyte developmental competence due its embryo-trophic effects and due to its protective effects against embryo-toxic agents [26,27,28]. To date, serum is still used to increase IVP efficiency in some bovine IVP laboratories both for research and commercial purposes. However, serum supplementation has been shown to have several disadvantages such as excessive lipid accumulation in the embryo, various alterations of embryo morphology, ultrastructure and kinetics of development [29], increased sanitary risks (including potential for Hepatitis B infections) and long-lasting epigenetic alterations, such as those leading to large offspring syndrome in cows [30,31,32,33]. Therefore, in human and some bovine IVP protocols, serum is replaced by other macro-molecules such as human or bovine serum albumin (BSA). Albumin acts as a carrier, provides substrates and growth factors and scavenges ROS [34].

In addition to albumin, insulin-transferrin-selenium (ITS) is commonly supplemented during routine serum-free bovine IVC due to the antioxidant properties of transferrin and selenium and for the mitogenic and anti-apoptotic effects of insulin [35,36,37,38]. Some commercially available human IVC media contain insulin [39]. 

In the present study, we hypothesized that supportive embryo culture conditions supplemented with anti-apoptotic, antioxidant and mitogenic factors such as serum or ITS can reduce oxidative stress levels and rescue the development and quality of embryos derived from metabolically compromised oocytes. 

To test this hypothesis, we used NEFA-exposure during bovine oocyte IVM as a model to mimic oocyte maturation under maternal metabolic stress conditions. Oocytes were exposed to either a high PA concentration (PA) or to a combination of high concentrations of NEFAs (PA, SA and OA; HCOMBI) compared to basal concentrations of NEFA (BASAL). After IVF, zygotes were cultured in SOF culture medium containing only bovine serum albumin (BSA) or supplemented with ITS or serum. We investigated the effects on developmental competence as well as on important embryo quality parameters: cell proliferation, apoptosis, embryonic cell metabolism and expression of a selected number of genes related to oxidative stress, mitochondrial unfolded protein response, endoplasmic reticulum stress and mitochondrial biogenesis.

## 2. Results

### 2.1. The Impact of High NEFA Supplementation during IVM

#### 2.1.1. Cumulus Cell Expansion

Cumulus cell expansion was significantly (*p* < 0.05) reduced in cumulus oocyte complexes (COCs) exposed to PA (1.90 ± 0.02) and HCOMBI (1.61 ± 0.02) compared with the BASAL control (physiological NEFA concentrations) (2.83 ± 0.001) [4].

#### 2.1.2. Embryo Development

A total of 3737 bovine COCs were matured in either BASAL, PA or HCOMBI NEFA concentrations during maturation and then cultured in BSA, ITS (+BSA) or serum. The effect of NEFAs on developmental competence was highly dependent on the culture condition, as shown in Table 1, Appendix A and described below. Supplementation of culture media with ITS or serum did not significantly influence cleavage and blastocyst rates of BASAL control oocytes when compared to IVC in BSA only (*p* > 0.1, see Appendix A for statistical comparison).

In basic BSA culture media, PA-exposure during IVM significantly reduced embryo cleavage rate, increased arrest at the 2-cell stage and reduced rates of development to the ≥4-cell stage at 48 h post-insemination (p.i.), and to the blastocyst stage at days 7 and 8 (*p* < 0.05) compared with BASAL control (Table 1). Culture of PA-exposed oocytes in serum-supplemented medium did not alleviate any of its negative impact on developmental competence when compared to the BASAL control group cultured in serum, as similar differences where observed. In contrast, development of PA-exposed oocytes was relatively improved with ITS supplementation where, despite lower 4 cell+ proportions, blastocyst rates on days 7 and 8 were similar to those observed in the BASAL-ITS control (*p* > 0.1) and significantly higher than those in the PA-BSA group (*p* < 0.05, see Appendix A). 

Exposure to HCOMBI NEFAs during IVM also reduced developmental competence in basic culture (BSA) compared to BASAL NEFAs, with increased arrest at the 2-cell stage (*p* < 0.1), lower proportions of 4 cell+ embryos (*p* < 0.05) and day 8 blastocysts (*p* < 0.1). This negative impact however was smaller compared with the effects seen after maturation in PA (Table 1). In contrast, supplementation of either ITS or serum during IVC alleviated the negative impact of HCOMBI exposure on embryo development as there was no difference anymore with the corresponding BASAL control groups (*p* > 0.1). 

### 2.2. Embryo Quality 

#### 2.2.1. Total Cell Count and Apoptotic Cell Index

The effect of BSA, ITS + BSA or serum supplementation during IVC on total cell numbers and apoptotic cell index (ACI) of blastocyst resulting from metabolically compromised oocytes was determined in 7 replicates using 498 embryos (Figure 1).

Overall, total cell counts of day 8 blastocysts were not influenced by PA or HCOMBI exposure in any of the IVC conditions compared to the corresponding BASAL controls (*p* > 0.1), and were not influenced by the ITS or serum supplementation either. As an exception, HCOMBI-ITS exhibited significantly higher total cell numbers compared to BASAL-ITS (137 ± 6 vs. 116 ± 5, *p* < 0.05) (Figure 1).

Apoptotic cell index (Figure 1) in day 8 BASAL control embryos was not significantly influenced by ITS and serum supplementation compared to BSA culture (*p* > 0.05). ACI in PA-BSA was similar to that in BASAL-BSA blastocysts (*p* > 0.1), however, ACI was significantly higher in PA-ITS and PA-Serum compared to BASAL-ITS (32.32 ± 5.2 vs. 13.43 ± 1.5, *p* < 0.05) and BASAL-Serum groups, respectively (29.34 ± 3.9 vs. 15.78 ± 1.6, *p* < 0.05). ACI of HCOMBI-derived blastocyst was similar to BASAL controls in BSA and serum culture, but it was significantly increased in ITS culture (24.31 ± 2.4 vs. 13.43 ± 1.5, *p* < 0.05).

#### 2.2.2. Metabolic Activity of Surviving Blastocysts

Metabolic assays were performed on 389 blastocysts (in individual culture from day 7 p.i. for 24 h) in four replicates. The morphological stage of each blastocyst was scored on days 7 and 8.

At the start of the metabolic assay, no significant differences were present when comparing day 7 blastocyst stages between the different treatment groups. The percentage of blastocysts that further developed from one morphological stage to another during the 24 h of single culture was similar among treatment groups (*p* > 0.05).

Within the BASAL control group, it was noticed that embryo culture in ITS-supplemented media did not influence blastocyst metabolic activity, while serum supplementation in embryo culture significantly increased lactate production (33.4 ± 2.1 vs. 22.2 ± 1.2 pmol/embryo/hour, Figure 2) and lactate:(2 glucose) ratio (1.31 ± 0.32 vs. 0.43 ± 0.07 pmol/embryo/hour) compared to the BASAL-BSA group (*p* < 0.05). 

In ITS-free and serum-free culture, PA-BSA blastocysts exhibited similar glucose and pyruvate consumption and lactate production compared to BASAL-BSA controls (*p* > 0.1). In contrast, HCOMBI-derived blastocysts consumed significantly less pyruvate under BSA culture compared to BASAL controls (11.1 ± 3.6 vs. 25.3 ± 2.4 pmol/embryo/hour, *p* < 0.05), while glucose and lactate metabolism were not influenced (*p* > 0.1).

In ITS culture, PA-derived blastocysts tended to consume lower pyruvate (17.2 ± 2.8 vs. 26.1 ± 2.6 pmol/embryo/hour, *p* < 0.1) and exhibited a significantly lower lactate:(2 glucose) ratio (0.20 ± 0.1 vs. 0.46 ± 0.03 pmol/embryo/hour, *p* < 0.05) compared to the BASAL-ITS group. Serum supplementation also reduced pyruvate consumption (14.9 ± 3.3 vs. 26.7 ± 2.9 pmol/embryo/hour) and lactate:(2 glucose) ratio (0.49 ± 0.1 vs. 1.31 ± 0.32 pmol/embryo/hour) of PA-derived embryos and significantly increased their glucose consumption (45.5 ± 6.0 vs. 27.8 ± 3.3 pmol/embryo/hour) compared to the BASAL-serum control group (*p* < 0.05). 

In contrast, HCOMBI-derived blastocysts consumed significantly less pyruvate under BSA culture compared to BASAL controls (11.1 ± 3.6 vs. 25.3 ± 2.4 pmol/embryo/hour, *p* < 0.05), while glucose and lactate metabolism were not influenced (*p* > 0.1). Furthermore, with ITS and serum supplementation during culture, the metabolic activity of HCOMBI blastocysts including pyruvate consumption were all similar to the corresponding BASAL controls (*p* > 0.1). HCOMBI-serum blastocysts consumed significantly less glucose (19.8 ± 1.7 pmol/embryo/hour) and displayed higher lactate:(2 glucose) ratio (1.11 ± 0.27 pmol/embryo/hour) compared to HCOMBI-BSA (35.6 ± 2.6, 0.39 ± 0.04 pmol/embryo/hour, respectively) and HCOMBI-ITS (32.5 ± 3.4, 0.57 ± 0.05 pmol/embryo/hour, respectively) blastocysts (*p* < 0.05). Pyruvate consumption was only higher when compared to HCOMBI-BSA blastocysts. 

#### 2.2.3. Gene Expression Analysis of Day 8 Blastocysts

Gene expression analysis was performed on a total of 7–16 blastocysts/pool/treatment, with 4 pools/treatment group. A general overview of the results is displayed in Appendix A.

PA-BSA blastocysts had a significantly higher expression of heat shock protein 60 (*HSP60*) (a marker of mitochondrial unfolded protein responses (UPR)) compared to BASAL-BSA (*p* < 0.05, Figure 3). This effect was completely alleviated in ITS- and serum-supplemented groups. PA-serum tended to have a lower expression of Catalase (antioxidant) compared to BASAL-serum (*p* < 0.1). The expression patterns of other genes of interest were not affected by PA-exposure in any of the culture conditions compared to BASAL controls. Within the PA-exposed group, embryos cultured in the presence of ITS displayed lower *HSP60* expression compared to those cultured in BSA (*p* < 0.1) or serum (*p* < 0.05). ITS supplementation also resulted in relatively higher mitochondrial transcription factor A(*TFAM)* expression (involved in mitochondrial biogenesis) in PA-ITS blastocysts compared to PA-BSA and PA-serum (*p* < 0.1).

On the other hand, exposure to HCOMBI did not influence the gene expression pattern in any of the culture conditions compared to the corresponding BASAL controls (*p* > 0.1). 

Regardless of the IVM condition, addition of serum during in vitro culture significantly reduced glutathione peroxidase (*GPx)* expression of embryos when compared to those cultured in the presence of either BSA, ITS (with BSA) or both.

## 3. Discussion

The aim of this study was to examine the effect of supportive culture conditions (ITS or serum supplementation) on development and quality of embryos derived from oocytes matured under lipotoxic conditions. We confirmed that a lipotoxic environment during IVM significantly reduced embryo developmental competence when cultured in basic SOF medium supplemented only with BSA. ITS-supplementation during in vitro culture, however, was able to alleviate the negative effects of a lipotoxic maturation environment on oocyte developmental competence. Nonetheless, surviving PA-derived blastocysts were overall lower in quality, as they displayed higher apoptotic cell indices, an aberrant cellular metabolism and few alterations in expression of genes involved in mitochondrial biogenesis and unfolded protein responses (UPRs). Addition of serum to the culture medium did not improve developmental competence of PA-exposed oocytes. However, similar to ITS, serum supplementation to PA-exposed embryos resulted in higher apoptotic cell indices and an aberrant cellular metabolism.

### 3.1. High NEFA Concentrations during IVM Affect Subsequent Embryo Development but Not Quality (Apoptosis, Cellular Metabolism and Gene Expression Analysis) When Cultured in Non-Supportive Conditions (Basic SOF-BSA Medium)

In the present study, maturing oocytes in both PA or HCOMBI reduced cumulus cell expansion compared to the BASAL-exposed COCs in all experiments, confirming what De Bie et al. [40] and Marei et al. [14] observed. Leroy et al. [4] linked the observation of poorly expanded COCs after maturation in the presence of PA to a high degree of late apoptotic and even necrotic cumulus cells. In line with these observations, COCs collected from obese patients have been shown to be inferior in quality with bad morphology and reduced cumulus cell expansion [7], which was linked to low maturation rates [41]. 

Oocytes exposed to elevated concentrations of PA during maturation and cultured in basic BSA showed significantly lower developmental competence compared to BASAL-exposed oocytes. This is in line with Paczkowski et al. [42], Aardema et al. [13] and Marei et al. [19]. HCOMBI-BSA embryos also displayed reduced developmental competence when compared to the BASAL control group, although to a lesser extent, with a relatively higher proportion of good quality (≥four-cell) embryos and higher blastocysts rates compared to PA-exposed oocytes. The fact that PA exposure had a more detrimental impact on embryo development compared to HCOMBI has been observed in previous studies in our laboratory also under ITS- and serum-free culture conditions [18,19,40]. As demonstrated in a human clinical study, PA concentration in the follicular fluid may increase without a concomitant increase in OA, which was associated with negative pregnancy results following ICSI [11]. The presence of OA, as a monounsaturated FFA, has been shown to partially protect the oocyte against the lipotoxic effects of saturated FFAs, such as PA and SA, leading to relatively higher embryo development rates [13]. OA also increases lipid storage, hereby compensating for the adverse effects of PA and SA by deviating the fatty acid overload of the mitochondria away towards lipogenesis and accumulation of lipid droplets [13]. 

After maturation in HCOMBI or PA and culture in BSA, we could not detect any reduction in quality of the surviving blastocysts compared to BASAL controls. Total cell numbers, ACI and embryo metabolism were similar to the controls. In addition, with the exception of higher *HSP60* expression (a marker of mitochondrial UPR), mRNA transcription of the tested endoplasmic reticulum (ER) stress, UPR, mitochondrial biogenesis and oxidative stress markers were also not affected. This might indicate that the cellular stress level present in these surviving blastocysts was relatively low as no other marks of oxidative stress or apoptosis were present. It may furthermore indicate that the embryos with high cell stress levels were arrested before reaching the blastocyst stage and only the best quality embryos survived. 

This is an important finding as it indicates that non-supported IVC conditions reduce the chance of a metabolically compromised oocyte to develop to the blastocyst stage. However, those that did develop further were of overall good quality based on the evaluated parameters. This notion is further confirmed when looking at the ITS and serum effects, as discussed later. 

It was noticed that the apoptotic cell index of BASAL-BSA control embryos was relatively higher (18.31% ACI) than what is expected under standard conditions. Therefore, as an extra validation, we compared BASAL NEFA exposure to FA-free and solvent (0.5% ethanol) IVM control groups (Appendix A). Embryo cleavage and blastocyst rates in both solvent and BASAL treatments during maturation showed no significant differences in developmental competence and total cell count when compared to control blastocysts. However, we noticed a significantly higher ACI (13.67% and 14.70% vs. 5.80%, respectively). These data suggest that the addition of ethanol to the in vitro maturation medium caused an increase in blastocyst apoptosis, explaining the high ACI in BASAL-exposed control oocytes. It has been shown previously that addition of ethanol to the IVM medium led to increased expression of caspase-3 [43].

### 3.2. Supplementation of IVC Medium with ITS Supports the Development of Lower Quality Oocytes That Otherwise Would Not Have Developed Further

It has been reported previously that development of embryos derived from oocytes matured under standard conditions could be significantly improved by ITS supplementation during IVC [27,28,44,45]. We could not confirm this in our BASAL matured oocytes as ITS supplementation did not result in increased developmental competence rates. However, our data showed that supplementation of ITS to the IVC medium of metabolically compromised oocytes (PA exposure) significantly improved developmental competence and yielded similar blastocyst rates compared to the BASAL controls. This shows that ITS, in contrast to IVC in basic media (only supplemented with BSA), supports the development of metabolically compromised oocytes that would not have cleaved or developed to a blastocyst in the absence of ITS. This effect was already visible when recording cleavage rates at 48 h p.i. (24 h after starting the ITS treatment). In addition, the ratio of blastocyst/cleaved embryos was significantly higher in PA-ITS- compared to PA-BSA-treated embryos, further confirming the supportive effect of ITS on oocyte developmental competence. 

Developmental competence is an important non-invasive assessment parameter indicating the ability of an oocyte to undergo successful cytoplasmic and nuclear maturation, fertilization and embryo development. In addition to developmental competence, it is also important to look at embryo quality as it determines further development [23]. When focusing on embryo quality, apoptotic cell indices in PA-ITS and HCOMBI-ITS embryos were significantly higher compared to BASAL-ITS-treated embryos. This does not support our hypothesis of the possible anti-apoptotic effect of insulin at this stage. Together with the results of oocyte developmental competence, this leads to a very important conclusion: ITS might be able to support the development of low-quality oocytes to the blastocyst stage that would have otherwise been arrested during earlier development. Although the produced embryo is morphologically normal, the high apoptotic rates put the subsequent developmental capacity after transfer of such embryo at risk. 

This notion was further supported when looking at embryo metabolism. PA-ITS embryos displayed lower pyruvate consumption and lower lactate:(2 glucose) ratio compared to BASAL-ITS embryos, which may indicate a decrease in Warburg metabolism. Warburg metabolism is a metabolic phenotype observed in healthy embryos during pre-implantation development allowing rapid cell proliferation and other important metabolic requirements like redox regulation and the production of biosynthetic molecules [46]. Lactate (a product from Warburg metabolism or aerobic glycolysis) production has been shown to be vital for embryo development as embryo development was reduced when the conversion from pyruvate to lactate was inhibited [47]. Therefore, reduced lactate production may contribute to the defective development of PA-derived embryos under ITS culture. Lactate production by the pre-implantation embryo also plays and important role in the first embryo-maternal cross-talk. It has been reported to facilitate several key functions during implantation, including biosynthesis and endometrial tissue breakdown [48].

Within ITS culture conditions, no significant differences were found when comparing the expression of genes related to mitochondrial UPRs and biogenesis, ER stress and oxidative stress of PA- and HCOMBI-derived blastocysts to the BASAL control group. This means that the significant increase in *HSP60* (a marker of UPR) detected in PA-blastocysts cultured in non-supplemented media (BSA) could be alleviated by ITS. This conclusion is further substantiated by a tendency to decrease HSP60 expression in PA-ITS embryos when compared to PA-BSA. 

Furthermore, PA-ITS embryos displayed an increased *TFAM* expression when compared to those cultured in BSA or serum, suggesting an increase in mitochondrial biogenesis. Khera et al. [49] reported that supplementation of selenium protects cells from mitochondrial stress through the upregulation of antioxidant systems and mitochondrial biogenesis. When matured in the presence of high NEFAs (PA or HCOMBI), embryos cultured in ITS-supplemented culture medium displayed significantly higher *GPx* expression when compared to those cultured in serum. ITS consists of selenium, which is known to promote *GPx* activity. Furthermore, it has been shown that ITS supplementation can significantly increase glutathione content in oocytes and display an insulin-like function to raise the developmental potential of oocytes [50]. These data may explain our observation that supplementing in vitro culture medium with ITS can support the development of lower quality oocytes that otherwise would not have developed further, by promoting its antioxidative activity in mitochondrial biogenesis.

### 3.3. Supplementation of IVC Medium with Serum Did Not Improve Developmental Competence and Quality of PA-Exposed Oocytes

Supplementation of serum to the culture medium did not further improve nor deteriorate developmental competence of PA-exposed oocytes when compared to PA-oocytes cultured in non-supplemented BSA media. In contrast to the effects of ITS supplementation, the addition of serum resulted in reduced developmental competence of PA-exposed oocytes when compared to BASAL-serum oocytes, hereby confirming earlier findings [15,51]. Furthermore, similar to the observations after culture in ITS medium, PA-exposed oocytes cultured in the presence of serum displayed significantly higher ACI compared to BASAL- and HCOMBI-exposed oocytes, indicating a decreased quality of the surviving blastocysts. Furthermore, glucose consumption was significantly higher in PA-serum blastocysts compared to BASAL- and HCOMBI-serum blastocysts, resulting in the observed decrease in the lactate:(2 glucose) ratio. This increase in glucose consumption is most likely due to an inhibition of the fatty acid oxidation (FAO) cycle, caused by the increased concentration of saturated PA [52]. This inhibition of the FAO has been shown to upregulate glucose metabolism, suggesting an adjustment towards oxidative phosphorylation to compensate for the loss of optimal ATP production via the FAO cycle [53,54]. Interestingly, glucose consumption of HCOMBI-serum blastocysts was significantly lower compared to PA-serum blastocysts and not significantly different from BASAL-serum controls. OA has been shown to increase the expression of genes linked to the FAO pathway. As a result, OA potentially accelerated the rate of complete FAO, hereby replacing the increased consumption of glucose [55].

It has been demonstrated before that gene expression in the developing embryo was changed when embryos were cultured in medium supplemented with serum [56]. In the present experiment, GPx (= antioxidant function) expression was significantly lower when embryos were cultured in serum compared to IVC in BSA or ITS, regardless of their exposure during in vitro maturation. Together with the other oocyte quality parameters, this decrease in GPx antioxidant gene expression confirms the reduced quality of the PA-serum surviving blastocysts. 

Interestingly, when morphologically good-quality blastocysts, resulting from PA-exposed oocytes cultured in serum, were transferred to healthy cows, surviving post-hatching embryos still showed decreased quality, indicated by growth retardation, altered metabolism and altered transcriptomic profile after 7 days of in vivo culture (14 days p.i.) when compared to the BASAL control group, indicating long-lasting effects [51]. Post-hatching development of HCOMBI embryos was not tested in this study. 

### 3.4. Supplementation of Serum to IVC Media of HCOMBI- and BASAL-Exposed Oocytes Alters Cellular Metabolism in Surviving Blastocysts

In our experimental set-up, supplementation of serum during IVC alleviated the negative impact of HCOMBI exposure on embryo development as there was no difference anymore with the corresponding BASAL control groups. With regards to embryo metabolism, HCOMBI-serum embryos displayed significantly lower glucose consumption compared to IVC in BSA or ITS. Furthermore, both pyruvate consumption and lactate:(2 glucose) ratio were increased, suggesting a preference towards Warburg metabolism in this group. BASAL-serum embryos displayed a significantly higher lactate production and lactate:(2 glucose) ratio when compared to IVC in the presence of BSA alone or ITS, also indicating a preference towards Warburg metabolism. These results may suggest that the addition of serum to the culture medium may support this metabolic feature. Warburg metabolism spares glucose from being used for energy production and shunts it towards other metabolic pathways such as the pentose phosphate pathway (PPP) [46]. During the PPP, many reducing equivalents are being produced. As such, we propose that the addition of serum leads to an increased production of reducing equivalents available for biosynthesis and production of reduced glutathione, a key intracellular antioxidant [48]. In line with this hypothesis, it was shown before that human hepatocellular carcinoma cells cultured in the presence of fetal bovine serum (FBS) typically display a Warburg-like metabolic profile. When replacing the FBS, metabolic analysis shows that the Warburg-like metabolic profile was restored to oxidative metabolic features of normal liver cells [57]. To summarize, BASAL- and HCOMBI-serum blastocysts displayed a preference towards Warburg metabolism, resulting in a possible increased production of reduced glutathione. Together with the displayed reduced GPx gene expression in these treatment groups, this might indicate that additional antioxidant defense might not be necessary, as those blastocysts displayed overall good quality.

## 4. Materials and Methods

All laboratory materials were purchased from Sigma-Aldrich (Overijse, Belgium) unless otherwise stated.

### 4.1. Experimental Design

Metabolically compromised oocytes were generated by exposure to different pathophysiological concentrations of NEFAs during IVM, a model that has been previously established and validated in our laboratory [15,16,58]. Subsequently, presumptive zygotes were cultured in vitro in different conditions: SOF media containing only BSA as a macromolecule, or BSA supplemented with ITS (as a mitogenic, anti-apoptotic and antioxidative supplement). This was compared to a group cultured in SOF media supplemented with serum (which naturally contains albumin and embryo-trophic factors but is less chemically defined). 

Bovine COCs were matured for 24 h in media supplemented with different concentrations of NEFAs, as measured in the FF of normal weight and obese women [10] and in cows during negative energy balance [4]: (1) A combination of basal concentrations of PA, SA, and OA (BASAL; 23, 28 and 21 µM, respectively; 72 µM total NEFA) as a physiological control, (2) high pathophysiological PA (150 µM) concentration together with basal SA and OA concentrations (PA) and (3) a combination of high, pathophysiological concentrations of PA, SA and OA (HCOMBI; 150, 75 and 200 µM, respectively; 425 µM total NEFA). 

After fertilization, presumptive zygotes from each maturation condition were cultured in modified synthetic oviductal fluid (mSOF) supplemented with either (1) 2% BSA, (2) 2% BSA and ITS containing 10 µg/mL insulin, 5.5 µg/mL transferrin and 6.7 ng/mL selenium (Fisher Scientific, Merelbeke, Belgium), or (3) 5% Fetal Bovine Serum (FBS), resulting in nine treatment groups in a 3 × 3 factorial design (see Figure 4).

Embryo developmental competence was assessed by recording cleavage rate (48 h p.i.) and blastocyst rate at day 7 and 8 p.i. (*n* = 3737). Blastocyst quality was assessed by examining total cell count and apoptotic cell index (*n* = 498). Furthermore, the metabolic profile (glucose and pyruvate consumption, lactate production; *n* = 389) and expression of a selected number of genes of interest (7–16 blastocysts/pool/treatment group) were also analyzed (see Figure 5). The number of replicates and COCs or blastocysts used for each outcome parameter are indicated in the results section. 

### 4.2. Preparation of NEFA Stocks and NEFA-Supplemented In Vitro Maturation Media

Stock solutions of SA, PA and OA were prepared in absolute ethanol. These stock solutions were added to maturation medium to obtain the final desired NEFA concentrations. Ethanol was adjusted to the same concentration (0.5%) in all treatment groups. Previous studies in our laboratory have shown that addition of ethanol (0.5%) or BASAL NEFA concentrations during IVM did not have any significant impact on embryo cleavage or development to the blastocyst stage compared to standard solvent-free and FFA-free (standard lab control) conditions when cultured in the presence of ITS [14] or serum [15]. We have also confirmed this in the presence of BSA alone (Appendix A). Therefore, based on these results, exposure to BASAL NEFA concentrations during IVM was used as a physiological control in the present study.

### 4.3. In Vitro Embryo Production Procedure 

#### 4.3.1. Oocyte Collection and In Vitro Maturation (IVM)

IVM of bovine oocytes was performed for 24 h as previously described by Desmet et al. [20]. Briefly, immature COCs surrounded by five or more compact cumulus cell layers (quality grade I) were retrieved from slaughterhouse ovaries and selected for serum-free IVM. The COCs were washed and cultured in 500 mL maturation medium supplemented with 20 ng/mL epidermal growth factor (EGF) stock II in groups of 50–60 COCs for 24 h in humidified air with 5% CO_2_ at 38.5 °C. Maturation medium consisted of TCM199 (Life Technologies, Merelbeke, Belgium) supplemented with 0.4 mM L-glutamine, 0.2 mM sodium pyruvate, 0.1 μM cysteamine and 50 μg/mL gentamicin. Maturation medium was supplemented with different concentrations of NEFAs, as mentioned in the experimental design.

#### 4.3.2. In Vitro Embryo Production

After maturation, COCs were in vitro fertilized using frozen-thawed semen of the same ejaculate from a proven-fertile bull. Straws were thawed in warm sterile water (37 °C for 30 s) and their content was centrifuged on a discontinuous Percoll^®^ gradient (90% and 45%, Amersham Biosciences, Roosendaal, The Netherlands) to select the viable, motile spermatozoa. COCs were co-incubated in groups of 100–120 in fertilization medium (Fert-TALP (Tyrode albumin lactate pyruvate) containing heparin as a capacitating agent) in a final concentration of 10^6^ sperm cells/mL for 22 h at 38.5 °C and 5% CO_2_ in a humidified incubator [4].

At 22 h of IVF, presumptive zygotes were denuded by vortexing (3 min). Zygotes were then washed and cultured in groups of 25 ± 4 in 75 µL mSOF medium supplemented with or without 2% BSA, ITS or 5% serum, as described in the experimental design. SOF-medium consisted of 2.77 mM myoinositol, 0.72 mM sodium pyruvate, 1.5% (*v*/*v*) minimum essential medium (MEM) 50X, 1% (*v*/*v*) non-essential amino acids (MEM-NEAA) 100X, 0.4 mM L-glutamine, 50 μg/mL gentamycin, 108 mM NaCl, 70 mM KCl, 120 mM KH_2_PO_4_, 10 mM MgSO_4_, 7 H_2_O, 60% lactate, 20 mM NaHCO_3_, 0.01% phenol red, 180 mM CaCl_2_, 2H_2_O and 10 mg/mL trisodium citrate. Culture plates were incubated at 38.5 °C, 5% CO_2_, 5% O_2_ and 90% N_2_ (maximum humidity) until day 7 or 8 p.i., according to the outcome parameter that was assessed [59].

### 4.4. Outcome Parameters

#### 4.4.1. Assessment of Cumulus Cell Expansion

As described by Marei et al. [60], cumulus cell expansion was evaluated following 24 h of IVM. Cumulus expansion was scored (0–3) using an Olympus SZX7 stereomicroscope: not expanded (score 0), poorly expanded (score 1), partially expanded (score 2) or fully (maximum) expanded (score 3) (Figure 6). An average score of all COCs was calculated for each treatment group.

#### 4.4.2. Embryo Developmental Competence

Total cleavage rate, two-cell block, number of embryos with ≥4-cells and fragmentation were recorded at 48 h p.i. Zygotes were categorized as fragmented if ≥20% of their cellular mass was fragmented [61]. Cleavage rate and blastocyst yield was assessed at days 7 and 8 p.i. using an inverted Olympus CKX41 microscope (Olympus, Aartselaar, Belgium). Blastocyst rates were presented as the number of blastocysts per total number of oocytes used.

#### 4.4.3. Blastocyst Energy Metabolism: Determination of Pyruvate and Glucose Uptake and Lactate Production

Assessment of embryo metabolism was performed on individual bovine blastocysts using an ultra-micro-fluorometric technique (the Tecan Infinite M200 spectrophotometer) as described in detail by De Bie et al. [40], a protocol adapted from Guerif et al. [62]. Briefly, individual day 7 embryos were incubated for 24 h in analysis medium (AM) with defined physiological concentrations of glucose and pyruvate. The AM resembled the standard SOF medium mentioned above but with 0.5 mM glucose, 0.4 mM pyruvate and no lactate. Day 7 blastocysts were cultured individually in 8 µl droplets of AM for 24 h under equilibrated mineral oil in 60 mm dishes at 38.5 °C, 5% CO_2_ and 5% O_2_ until day 8. A few droplets were left without embryos and were used as blanks. Blastocyst morphological stage was scored at the start and end of the metabolic assay. Blastocysts were then removed and the dishes containing AM droplets were sealed and stored at −80 °C until analysis. 

After removal, the blastocysts were washed in phosphate-buffered saline with polyvinylpyrrolidone (PBS-PVP) and transferred individually to a labelled 96-well plate containing paraformaldehyde (PFA) 4% for fixation (20 min), then washed and stored at 4 °C for differential staining to determine their quality (see below). 

Absolute concentrations of glucose, pyruvate and lactate in spent media droplets were determined using enzymatic reactions and standard curves (0–0.5 mM), as described by Guerif et al. [62]. To determine the rate of consumption or production for each metabolite by each embryo, we calculated the difference in metabolite concentration in its spent medium droplet compared to that in the blank droplets. All samples were measured in duplicate. Lactate production and glucose and pyruvate consumption were expressed as pmol/embryo/h. In addition, lactate:(2 glucose) ratios (1 mol glucose produces 2 mol lactate) were calculated to estimate the metabolic pathway by which glucose was preferentially metabolized. For each measure, the coefficient of variance (CV) was monitored: glucose displayed an intra- and inter-assay CV of 7.0% and 7.0% respectively, while CV’s of lactate were 8.4% and 8.2% and CV’s of pyruvate were 7.8% and 7.9%, respectively.

#### 4.4.4. Assessment of Blastocyst Cell Number and Apoptotic Cell Index

Fixed blastocysts were immuno-stained with anti-cleaved caspase-3 antibody (Asp 175; Cell Signaling Technology) to determine the number of apoptotic cells, and all nuclei were counter-stained with Hoechst for total cell counting. Briefly, fixed day 8 blastocysts were permeabilized overnight at 4 °C using 1% Triton X-100 and 0.05% Tween 20 in PBS. Blocking was performed for 2 h at 4 °C in 10% normal goat serum in 0.05% Tween 20-PBS. Caspase-3 labelling was done using a rabbit anti-cleaved caspase-3 (1:250) in blocking solution overnight at 4 °C. Embryos were then washed in PBS-PVP and incubated with the secondary goat anti-rabbit fluorescein isothiocyanate (FITC) antibody (1:200) in blocking solution. Counter-staining was performed using PBS-PVP containing Hoechst 33342 (50 µg/mL) for 10 min at room temperature. After staining, the blastocysts were washed in PBS-PVP and mounted on slides in a drop of 1% 1,4-diazabicyclo[2.2.2]octane (DABCO) (in 90% glycerol and 10% PBS). The blastocysts were examined under a fluorescence microscope (Olympus IX71, X-cite series 120 Q) with 4′,6-diamidino-2-phenylindole (DAPI) and FITC filters at 200× magnification and images were acquired using Cell Sens-Standard software at the same exposure and gain settings. Total cells and caspase-3-positive cells were counted. 

#### 4.4.5. Blastocyst RNA Extraction, Reverse Transcription and Quantification of Gene Expression by Quantitative Polymerase Chain Reaction (qPCR)

Pools of at least ten day 8 blastocysts from each treatment were washed and transferred to a 1.5 mL vial in minimal volume of 0.1% PBS-PVP and snap-frozen and stored at −80 °C until further processing.

Total RNA from each replicate was extracted and purified using the PicoPureTM RNA Isolation Kit (Thermo Fisher Scientific, Asse, Belgium). RNA was isolated following the Manufacturer’s instructions, with minor modifications. Extracted RNA was treated with DNase (Qiagen, Venlo, The Netherlands). The RNA concentration and purity were checked using a BioAnalyzer (Agilent). After extraction, cDNA synthesis was performed using a Sensiscript reverse transcriptase (RT) kit as described by Marei et al. [19].

Gene transcripts were quantified by quantitative Polymerase Chain Reaction (qPCR) using SYBR green (SsoAdvanced SYBR Green supermix, Bio-Rad, Temse, Belgium). All samples were analyzed in duplicate. Quantification was normalized using the geometric mean of three housekeeping genes: *18 S*, *H2AFZ* and *YWHAZ*, calculated by geNorm software (geNorm, Camberley, UK). The comparative quantification cycle (Cq) method, ‘2–ΔΔCq’, was used to quantify the relative expression level of each gene, as described by Livak and Schmittgen [63]. Fold-changes of all studied genes were calculated compared to the control as a reference (=1 fold-change) and expressed as fold-change ± standard error of the mean (S.E.M.).

Transcript abundance of genes involved in mitochondrial unfolded protein responses (*HSP10* (official name: *HSPE1)* and *HSP60* (official name: *HSPD1)*), oxidative stress (*GPx*, catalase (*CAT*) and superoxide dismutase 2 (*SOD2*)), endoplasmic reticulum stress (*ATF4* and *ATF6*) and mitochondrial biogenesis (*TFAM*) were analyzed. Primer details are shown in Appendix A.

### 4.5. Statistical Analysis

Statistical analysis was performed using IBM Statistics SPSS 25. Categorical data of cleavage and blastocyst rates were compared between the same maturation or culture groups using a binary logistic regression model. Interaction between treatment (fixed factor) and repeat (random factor) effects was evaluated, and since they were not significant, this interaction term was omitted from the final model. Numerical data of total cell numbers, apoptotic cell index, gene expression, glucose and pyruvate consumption and lactate production of blastocysts were compared between relevant groups for equality of variance and normality of distribution and analyzed using a linear mixed model. When equal variance could not be assumed, a non-parametric Kruskal–Wallis and Mann–Whitney test were performed for each comparison. A post hoc test was performed for multiple comparison with a Bonferroni correction. Data are presented as mean ± S.E.M. *p*-values < 0.05 are considered significant.

## 5. Conclusions

We confirmed that high NEFA exposure during IVM significantly reduced embryo developmental competence when cultured in basic SOF medium (supplemented only with BSA). ITS supplementation during in vitro culture of PA-exposed oocytes supported the development of lower quality embryos during earlier development. However, this increase in blastocyst yields (which may be due to supporting defective embryos to develop) had negative consequences on the quality (expressed as apoptotic cell index, cellular metabolism and gene expression analysis) of the surviving blastocysts. In contrast, addition of serum to the culture medium did not significantly improve developmental competence of PA-exposed oocytes. Furthermore, serum supplementation to PA-exposed embryos resulted in higher apoptotic cell indices and an aberrant cellular metabolism.

Based on these data, we conclude that some of the routinely used supportive embryo culture supplements like ITS and serum may increase IVF success rates of metabolically compromised oocytes but this may increase the risk of reduced embryo quality and may thus have other long-term consequences on establishment of pregnancy or on offspring health. These results highlight the importance of embryo quality assessment and long-term follow-up of developmental programming and postnatal health before approving new applications of any in vitro supportive treatments in IVF clinics. Further research is required to unravel this.

## Figures and Tables

**Figure 1 ijms-21-08206-f001:**
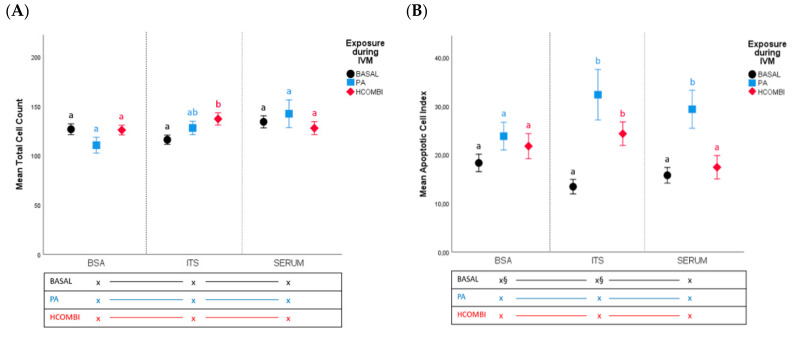
Mean total cell count (**A**) or apoptotic cell index (**B**) of Day 8 blastocysts resulting from oocytes exposed to BASAL (physiological NEFA concentrations), PA and HCOMBI NEFAs during maturation and cultured in the presence or absence of bovine serum albumin (BSA), ITS or serum. Data are presented as mean percentage ± SEM. Vertical superscripts (a, b) indicate significant differences between maturation conditions within the same IVC condition (*p* < 0.05). Horizontal superscripts (xindicate significant difference between culture conditions within the same in vitro maturation (IVM) condition (*p* < 0.05). Values labeled with “§” tend to be different from each other at *p* < 0.1 and > 0.05.

**Figure 2 ijms-21-08206-f002:**
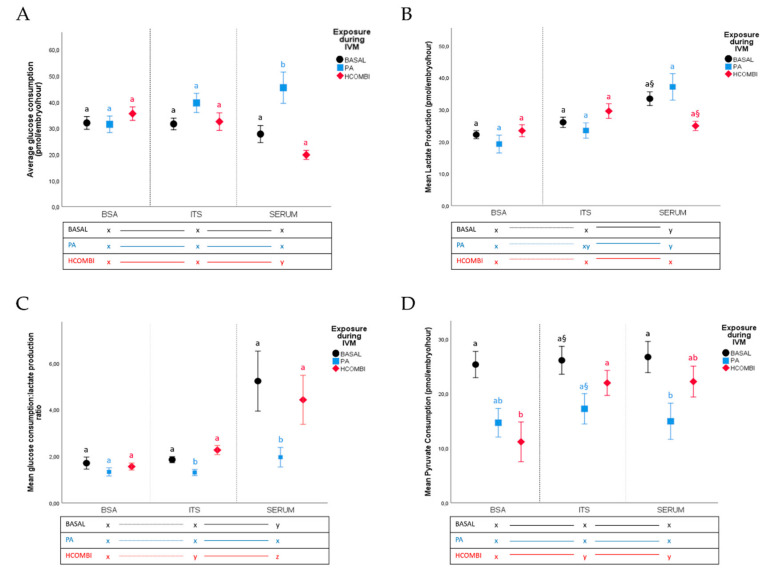
The effect of BSA, ITS or serum supplementation during IVC on (**A**) average glucose consumption, (**B**) lactate production, (**C**) lactate:(2 glucose) ratio and (**D**) pyruvate consumption (pmol/embryo/hour) of blastocysts resulting from BASAL-, PA- or HCOMBI-exposed oocytes. Data are presented as mean percentage ± SEM. Vertical superscripts (a, b) indicate significant differences between maturation conditions within the same IVC condition (*p* < 0.05). Horizontal superscripts (x, y, z) indicate significant difference between culture conditions within the same IVM condition (*p* < 0.05). Values labeled with “§” tend to be different from each other at *p* < 0.1 and > 0.05.

**Figure 3 ijms-21-08206-f003:**
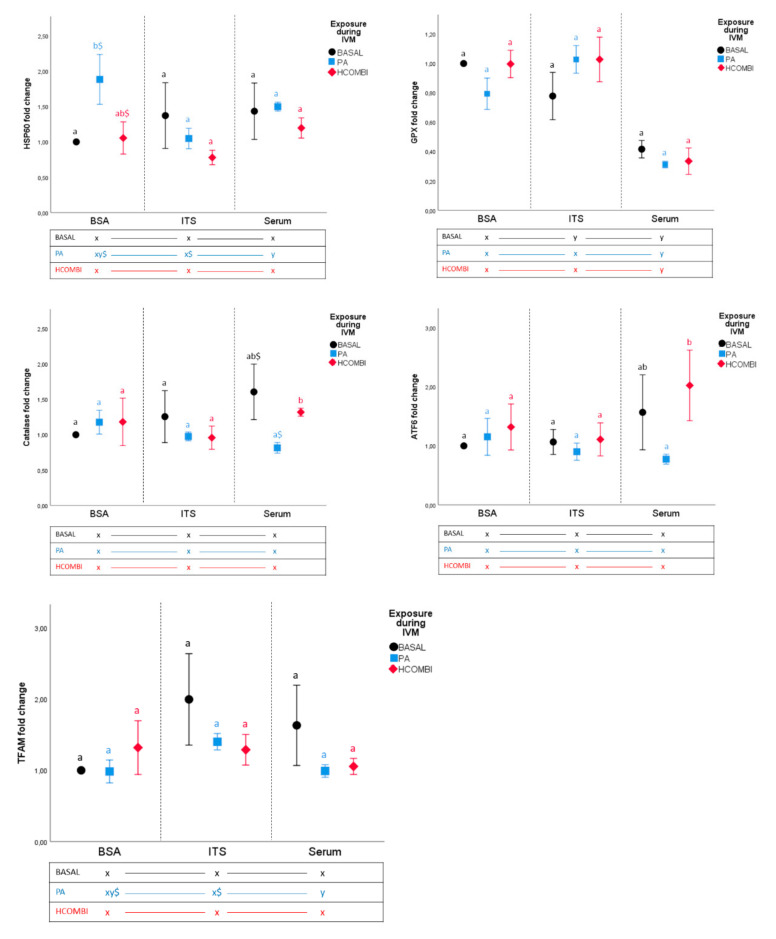
mRNA expression (fold change) of genes related to cellular stress of blastocysts resulting from BASAL-, PA- or HCOMBI-exposed oocytes cultured in serum, BSA and/or ITS. Data are presented as fold change ± SEM. Vertical superscripts (a, b) indicate significant differences between maturation conditions within the same IVC condition (*p* < 0.05). Horizontal superscripts (x, y) indicate significant difference between culture conditions within the same IVM condition (*p* < 0.05). Values labeled with “$” tend to be different from each other at *p* < 0.1 and > 0.05.

**Figure 4 ijms-21-08206-f004:**
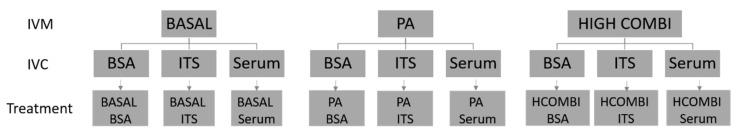
Overview of the nine treatment groups: BASAL-BSA, BASAL-ITS, BASAL-serum, PA-BSA, PA-ITS, PA-serum, HCOMBI-BSA, HCOMBI-ITS, HCOMBI-serum.

**Figure 5 ijms-21-08206-f005:**
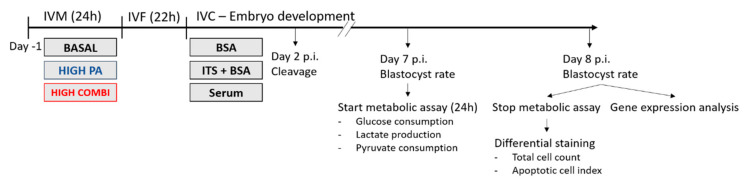
Overview of the experimental design.

**Figure 6 ijms-21-08206-f006:**
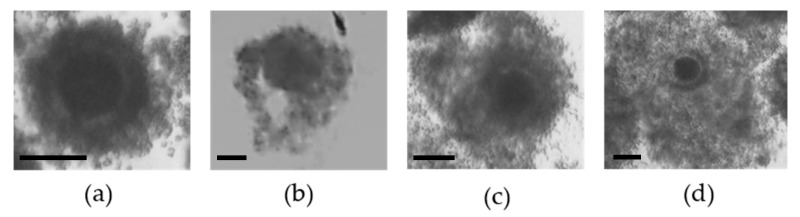
Grades of cumulus cell expansion in cumulus-oocyte complexes (COCs) after 24 h incubation in maturation media. (**a**) unexpanded COC (score 0), (**b**) poorly expanded COC (score 1), (**c**) partially expanded COC (score 2) and (**d**) and full y expanded COC (score 3). Scale bar = 100 µm.

**Table 1 ijms-21-08206-t001:** The effect of insulin-transferrin-selenium (ITS) or serum supplementation during in vitro culture (IVC) on the developmental competence of oocytes matured under metabolic stress conditions (PA (high palmitic acid) or HCOMBI NEFAs (a combination of pathophysiological, high concentrations of non-esterified fatty acids).

Maturation	Culture	Oocytes Used	Cleaved Embryos	Two-Cell Stage	Four-Cell Plus	Fragmented Embryos	Day 7 Blastocysts	Day 8 Blastocysts
BASAL	BSA	515	395 ^a^ (76.69%)	48 ^a§^ (9.32%)	255 ^a^ (49.51%)	63 (12.23%)	105 ^a^ (20.38%)	125 ^a§^ (24.27%)
PA	BSA	270	176 ^b^ (65.18%)	42 ^b^ (15.55%)	81 ^b^ (30.00%)	35 (12.96%)	30 ^b§^ (11.11%)	37 ^b^ (13.70%)
HCOMBI	BSA	527	377 ^ab^ (71.53%)	63 ^ab§^ (11.95%)	221 ^c^ (41.93%)	65 (12.33%)	87 ^ab§^ (16.50%)	96 ^ab§^ (18.21%)
BASAL	ITS	510	406 ^§$^ (79.6%)	63 (12.35%)	250 ^x^ (49.01%)	56^§^ (10.98%)	124 (24.31%)	139 (27.25%)
PA	ITS	278	200 ^$^ (71.94%)	43 (15.46%)	110 ^y^ (39.56%)	30 (10.79%)	52 (18.7%)	68 (24.46%)
HCOMBI	ITS	525	387 ^§^ (73.71%)	46 (8.76%)	235 ^xy^ (44.76%)	84 ^§^ (16.00%)	101 (19.23%)	120 (22.85%)
BASAL	Serum	488	348 ^jk§^ (71.31%)	73 ^j^ (14.95%)	182 ^j^ (37.29%)	55 (11.27%)	119 ^j^ (24.39%)	136 ^j^ (27.86%)
PA	Serum	247	156 ^j§^ (63.15%)	54 ^k^ (21.86%)	53 ^k^ (21.45%)	30 (12.14%)	30 ^k^ (12.14%)	45 ^k^ (18.21%)
HCOMBI	Serum	377	275 ^k^ (72.94%)	55 ^jk^ (14.58%)	149 ^j^ (39.52%)	41 (10.87%)	88 ^j^ (23.34%)	91 ^jk^ (24.13%)

% values refer to proportions from total number of the used oocytes. Values with different superscripts within the same in vitro culture group and column are significantly different at *p* < 0.05. Values labeled with a, b and c are used within BSA IVC; x, y is used for culture in the presence of ITS; j, k is used within Serum IVC. Values labeled with “§” or “$” tend to be different from each other at *p* < 0.1 and > 0.05.

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
