# Peer review of "Rescue Potential of Supportive Embryo Culture Conditions on Bovine Embryos Derived from Metabolically Compromised Oocytes"

_ijms, 2020, doi:10.3390/ijms21218206_

Round 1
Reviewer 1 Report
Commentary peer review: Rescue potential of supportive embryo culture conditions on bovine embryos derived from metabolically-compromised oocytes.
Introduction
L40 - 41
When FF of obese patients was added during bovine in vitro oocyte maturation (IVM), oocyte developmental competence and embryo quality were negatively affected [9].
Is supplementation of BASAL concentration of NEFA vital for bovine blastocyst development? Does maturation media TCM199 without BASAL NEFA produce bovine blastocyst as well?
Results
Table 1
Perhaps it would be good to simply show fertilization rate, blastocyst rate day 7 (if i understand well that day 8 blastocyst is post embryo metabolism analysis) and fragmentation rate for respective results.
L130 – 135
Exposure to HCOMBI NEFAs during IVM also reduced developmental competence in basic culture (BSA) compared to BASAL NEFAs with increased arrest at the 2-cell stage (P<0.1), lower proportions of 4cell+ embryos (P<0.05) and day 8 blastocysts (P<0.1). This negative impact however was smaller compared with the effects seen after maturation in PA (table 1). In contrast, supplementation of either ITS or serum during IVC alleviated the negative impact of HCOMBI exposure on embryo development as there was no difference anymore with the corresponding BASAL control groups
Is addition of serum then more physiological? As you used FBS have you considered effects of internal gonadotropins in the serum on the embryo development?
Figure 1-2
I do not clearly see the difference in significance, way how is presented, could you please consider amend the visual interpretation of figures 1-3
Figure 3
Perhaps mRNA expression would be more convenient to show in form of a table with fold change in log2 values and show respective P values
Discussion
L375 – 378
PA-exposed oocytes cultured in serum, were transferred to healthy cows, surviving post-hatching embryos still showed decreased quality, indicated by growth retardation, altered metabolism and altered transcriptomic profile after 7 days of in vivo culture (14 days p.i.),
Do you have such in vivo data for BASAL and HCOMBI serum as well?
L386-389
BASAL-serum embryos displayed a significantly higher lactate production and lactate :(2glucose) ratio when compared to IVC in the presence of BSA alone or ITS, also indicating a preference towards Warburg metabolism. These results may suggest that the addition of serum to the culture medium may support this metabolic feature.
L392 - 393
we propose that the addition of serum leads to an increased production of reducing equivalents available for biosynthesis and production of reduced glutathione, a key intracellular antioxidant
Do you have some data to support GSH production?
L398 - 400
BASAL- and HCOMBI-serum blastocysts displayed a preference towards Warburg metabolism resulting in a possible increased production of reduced glutathione. Together with the displayed reduced GPx gene expression in these treatment groups,
Will you be able to confirm GSH and GPX change in expression by Western Blot as well?
Materials and Methods
L415
NEFA measured in the FF of obese women [9] and in cows during negative energy balance [4]:
Have you obtained the NEFA data in the FF for normal BMI women?
L437 – 439
Previous studies in our laboratory have shown that addition of ethanol (0.5%) or BASAL NEFA concentrations during IVM did not have any significant impact on embryo cleavage or development to the blastocyst stage
>> 284 - 285 Therefore, as an extra validation, we compared BASAL NEFA exposure to FA-free and solvent (0,5% ethanol) IVM control groups (data not shown).
It might be good to show your blastocyst development data for IVM media with 0.5% ethanol only, at least in Table 1
L456
After maturation, COCs were in vitro fertilized for 22h using frozen semen of a bull with proven in vitro fertility, as described by Leroy, et al. [4].
How do you account for the quality of the frozen semen?
Could you please briefly describe sperm thawing and capacitation if possible?
L458
Zygotes were then washed and cultured in groups of 25±4 in 75 μl mSOF medium
Is 75 ul sufficient amount of media for 25 zygotes for 7 days? Has media been exchanged during the cultivation? Has the cultivation been done under oil? If so, please state what oil?
L463
Culture plates were incubated at 38.5°C, 5% CO2, 5% O2, and 90% N2 (maximum humidity) until day 7 or 8 p.i.,
I can see your embryos have been cultivated in hypoxic conditions, do you think displayed preference towards Warburg metabolism is mostly caused by BASAL- and HCOMBI-serum? Is 5% O2 for entire embryo culture physiological, if so please add supportive references?
„Day 5 compacting bovine embryos were cultured under different oxygen tensions (2%, 7%, 20%) and the effect on the expression of oxygen-regulated genes, development, and cell number allocation and HIFalpha protein localization were examined. Bovine in vitro-produced embryos responded to variations in oxygen concentration by altering gene expression“ Harvey et al., Oxygen-regulated gene expression in bovine blastocysts, BOR 2004
L468 - 469
Cumulus expansion was scored (0-3) using an Olympus SZX7 stereomicroscope: not expanded (score 0), poorly expanded (score 1), partially expanded (score 2) or fully (maximum) expanded (score 3).
Please provide some pictures of COC expansion and grading in the results section.
L535 – 536
Transcript abundance of genes involved in mitochondrial unfolded protein responses (HSP10 and HSP 60), oxidative stress (GPx, CAT en SOD2), endoplasmic reticulum stress (Atf4 and Atf6) and mitochondrial biogenesis (TFAM)
Do you also have 7 day blastocyst data to compare?
Do you have any Western Blot data confirming changes in expression level upon treatment for the candidate genes?
Would be nice to see how expression of HIF-1 alpha is behaving.
Reviewer 2 Report
In summary, this manuscript tests the hypothesis that metabolically-compromised oocytes (via exposure to PA or HCOMBI during IVM) may be improved during embryo culture if conditions are supplemented with ITS or serum (due to anti-oxidant and anti-apoptotic conditions).
Overall strengths and weaknesses
Strengths:
- Well-designed study with clear methodology and appropriate biologic plausibility with hypotheses.
- Organized results with figures that accurately reflect data.
- Results are meaningful and have the potential to result in future, translational research that could improve human IVF culture systems.
Weaknesses:
- The overall conclusion that potentially abnormal embryos are supported to the blastocyst stage of development with supplemental culture conditions would best be assess with offspring data, and this is lacking. PGT-A data could also be a helpful adjunct in future studies.
Specific comments on individual sections:
Introduction
- Comprehensive, with appropriate references included. Well done.
Results
- During IVM, can you provide the duration of oocyte NEFA-exposure? You referenced the previously-established protocol for IVM but I think this information is directly applicable to your hypothesis and should be included. Do we know what proportion of oocytes were GV versus MI prior to IVM? Could they have had differential outcomes with regard to developmental competence? Was the duration of NEFA-exposure different for any of the cohorts of oocytes? And was the maturity rate different among the NEFA exposed cohorts?
- During embryo development, were there observations related to abnormal cellular division in the early cleavage stage? For example, one cell to three cell?
- Figures: Consider removing the lines used in the figures. Though you state that the variables are not continuous, it is visually difficult not to see that relationship in the figures.
- Can you define HCOMBI? HIGH COMBI mean? And BASAL?
- Biostatistical analyses were appropriate.
Discussion
- High NEFA affected embryo development but not “quality” (line 247) – quality is yet to be determined with unknown genetics of embryos and unknown offspring effects. Please consider changing “but not quality” to specific parameters that were assessed to observe characteristics that are potentially compatible with high-quality embryos. Mean total cell count and blastocyst development in human embryos is not always linearly related to genetic euploidy.
Conclusions
- The conclusion that NEFA exposure during IVM significantly reduced embryo developmental competence is an appropriate assessment of the presented data. ITS supplementation during in vitro culture of PA-exposed oocytes potentially support the notion that lower quality embryos are able to survive to blast but may otherwise have undergone apoptosis. This latter conclusion may be overstated as direct evidence of embryo quality (offspring health, genetic ploidy of blastocyst) is not available.
